# μ-opioid receptor system mediates reward processing in humans

Lauri Nummenmaa[1,2], Tiina Saanijoki[1], Lauri Tuominen[1], Jussi Hirvonen[1,3], Jetro J. Tuulari[1], Pirjo Nuutila [1] & Kari Kalliokoski[1]

The endogenous μ-opioid receptor (MOR) system regulates motivational and hedonic processing. We tested directly whether individual differences in MOR are associated with neural reward responses to food pictures in humans. We scanned 33 non-obese individuals with positron emission tomography (PET) using the MOR-specific radioligand [$^{11}$C]carfentanil. During a functional magnetic resonance imaging (fMRI) scan, the subjects viewed pictures of appetizing versus bland foods to elicit reward responses. MOR availability was measured in key components of the reward and emotion circuits and used to predict BOLD-fMRI responses to foods. Viewing palatable versus bland foods activates regions involved in homeostatic and reward processing, such as amygdala, ventral striatum, and hypothalamus. MOR availability in the reward and emotion circuit is negatively associated with the fMRI reward responses. Variation in MOR availability may explain why some people feel an urge to eat when encountering food cues, increasing risk for weight gain and obesity.

[1] Turku PET Centre, University of Turku and Turku University Hospital, 20520 Turku, Finland. [2] Department of Psychology, University of Turku, 20014 Turku, Finland. [3] Department of Radiology, University of Turku, 20014 Turku, Finland. Correspondence and requests for materials should be addressed to L.N. (email: lauri.nummenmaa@utu.fi)

The human reward system promotes motivated behavior toward signals providing safety and opportunities for homeostasis. The endogenous opioid system is intimately involved in both hedonic functions and incentive motivation, and also in generating pleasurable sensations when consuming palatable foods[1,2]. Injection of μ-opioids into the mesolimbic reward system is rewarding in its own right[3], and μ-opioids receptor (MOR) stimulation in the shell of nucleus accumbens increases pleasure obtained from foods and may also promote feeding[4]. Similarly, opioid agonists increase and opioid antagonist decrease food intake and hedonic responses to palatable foods, respectively[5–8], whereas inverse MOR agonists diminish hedonic impact of feeding[9]. These data are paralleled by human imaging work showing that feeding triggers endogenous opioid release in the brain's reward circuit[10,11], possibly contributing to pathophysiology of obesity. Repeated overstimulation of the MOR system following food consumption may lead to lasting downregulation of MOR, thus providing a candidate neurobiological mechanism that reinforces overeating in obese individuals. Indeed, positron emission tomography (PET) studies have established that MOR levels are downregulated in patients with morbid obesity and binge-eating disorder[10,12–14].

In addition to homeostatic signaling, feeding involves preferences and habits that have been established by repeated reinforcing rewards. Numerous functional magnetic resonance imaging (fMRI) studies have established that merely viewing food-related cues engage the brain's reward circuit[15–17], possibly promoting food-seeking behavior. Indeed both obese subjects[18–20] and individuals with high genetic risk for obesity[21] show increased reward circuit responses when seeing pictures of foods. Individual differences in reward drive are also associated with both cerebral MOR availability[22] and reward circuit responses to food cues[15]. Because MOR downregulation is a hallmark of overeating and obesity[10,12,13], individual differences in MOR system could mediate reward responses and concomitant urges to eat when encountering food cues, such as pictures in advertisements. However, this hypothesis currently lacks direct empirical support.

Here we show that individual differences in limbic and frontocortical MOR availability are associated with reward circuits' responses to visual food cues in healthy non-obese individuals. We measured brain MOR availability using PET with [11C]carfentanil, a specific radioligand for MOR, and extracted the availability of MOR in key components of the reward and emotion circuit, as well as globally in in the brain. We then had participants undergo an fMRI scan where they viewed pictures of appetizing and bland foods (Fig. 1), while their attention was focused away from the hedonic and motivational aspects of the pictures. Based on prior observations on amplified reward responses to food cues in the obese[18–20], endogenous opioid release following feeding[11] and lower MOR density as a compensatory desensitization phenomenon due to overstimulation of MOR by endogenous opioids[10,12,13], we expected to see largest blood-oxygen-level-dependent (BOLD)-fMRI reward responses in subjects with lowest MOR availability. We show that such implicit reward processing activates frontal, striatal and limbic components of the reward circuit, and that the magnitude of the anticipatory reward responses is linearly associated with the MOR availability in the reward circuit. Such variation in MOR-dependent reward responsiveness may explain why some people feel an urge to eat when encountering food cues, increasing risk for weight gain and obesity.

## Results

**Regional effects in PET and fMRI.** Figure 2 shows mean [11C] carfentanil binding across the whole sample. Full-volume analysis did not yield significant associations between MOR availability and BMI. However, ROI analysis (Fig. 3) revealed a negative association in the amygdala ($r = -0.39$, $p < 0.05$), consistent with previous studies in obese patients[12]. In one-way testing, significant association was also observed in thalamus, putamen, and ventral striatum ($rs < -0.29$, $ps < 0.05$). Viewing appetizing versus bland foods activated the reward and emotion circuits reliably (Fig. 4 top row). Significant activation clusters were observed in the amygdala and hippocampus, ventral striatum, hypothalamus, and thalamus. Additional activations were observed in the precuneus, posterior cingulum, and occipital and ventral visual cortices. These effects were independent of the subjects BMI. Essentially, a similar pattern was observed in complementary permutation-based nonparametric statistical testing. The results remained essentially unchanged when age and pre-scan hunger levels were included as nuisance covariates in the analysis.

**Fusion analysis of PET-fMRI data.** We next tested whether regional MOR availability would be associated with the visual

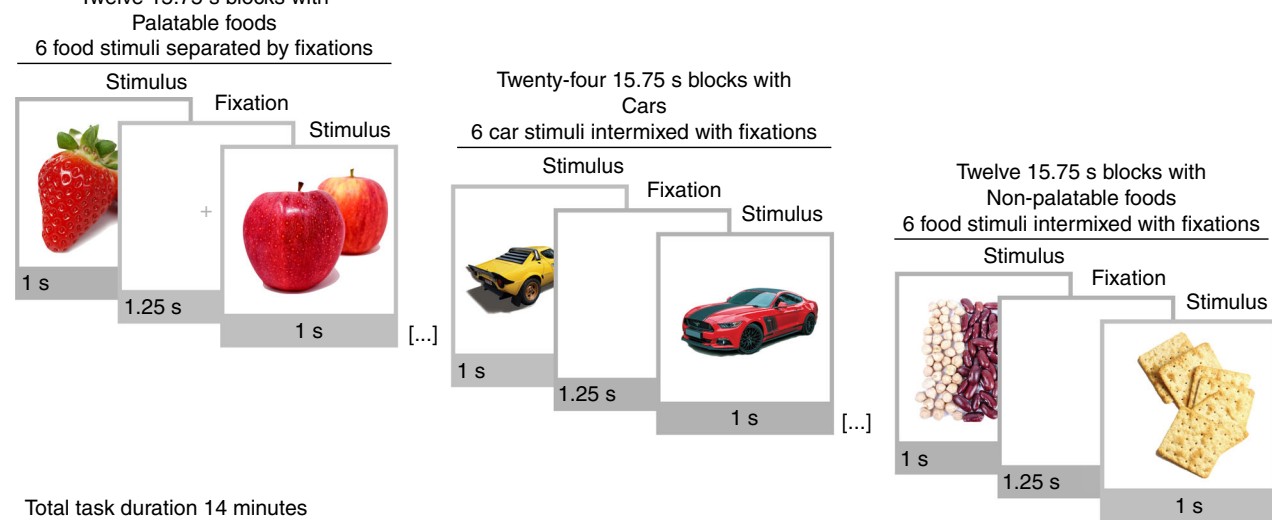

**Fig. 1** Experimental design for fMRI. Subjects viewed alternating 15.75-s epochs with palatable foods, non-palatable foods or cars. Each block contained six stimuli from one category, intermixed with fixation crosses

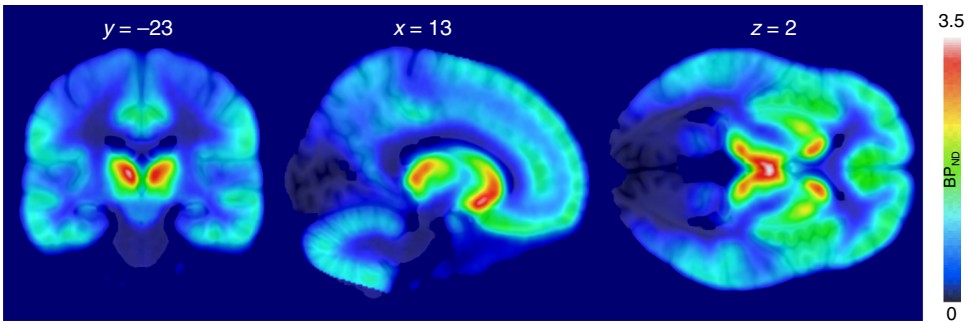

**Fig. 2** Mean distribution of MOR in the brain as measured with [$^{11}$C]carfentanil ($n = 33$) showing widespread MOR expression with highest densities in striatum, thalamus, and cingulate cortex

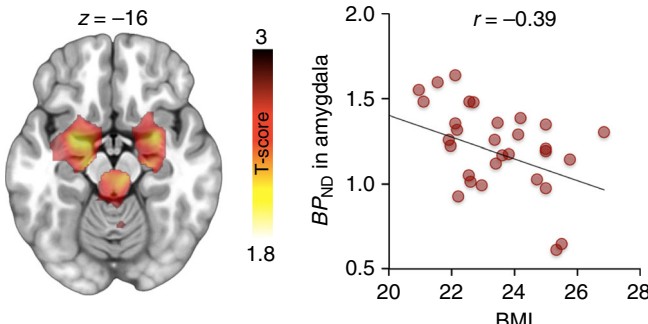

**Fig. 3** Association between BMI and MOR availability in the amygdala ($n = 33$). The full-volume data are shown at $p < 0.05$ uncorrected for visualization purposes

reward responses measured with fMRI. To that end, BOLD responses to the appetizing minus bland foods were predicted with regional MOR availabilities in full-volume SPM analysis. We found that, in general, MOR availability in the tested ROIs were associated with BOLD-fMRI reward responses in the ventral striatum, amygdala, hippocampus, orbitofrontal, frontal, and cingulate cortices, as well as somatosensory cortices (Fig. 4 bottom row and Fig. 5). The overall pattern of results was similar for all the tested ROIs with the exception that the $BP_{ND}$ in the caudate was not associated with BOLD-fMRI responses in any brain region, and that the frontocortical BOLD responses were only associated with $BP_{ND}$'s in the amygdala, putamen, thalamus, and ventral striatum (Fig. 6). Results using the global MOR availability closely mirrored these findings (data not shown). The overall distribution of the effects was similar in nonparametric testing, yet only associations between the $BP_{ND}$ in thalamus and BOLD responses exceeded significance.

Finally, to ensure that the association between MOR availability and responses to appetizing versus bland foods were specific to processing the reward value of the foods, we computed the contrast between all foods (appetizing and bland foods) versus car pictures, and predicted the resulting contrast with regional MOR availabilities as detailed above. Overall, viewing foods versus car pictures elicited clearly differential activation patterns in the occipito-temporal, parietal, and frontal cortices. However, these overall responses in these or other regions were not statistically significantly associated with MOR availability in any tested region.

## Discussion

Our main finding was that endogenous MOR availability in the reward and emotion circuit is associated negatively with BOLD-fMRI reward responses when viewing appetizing versus bland foods. That is, lower baseline MOR availability was associated with stronger BOLD responses to appetizing versus bland foods in frontal and cingulate cortices, as well as in striatum and amygdala. This effect was not contingent on the subjects' BMI. These data suggest that individual variation in MOR expression may explain why some people cannot resist the urge to eat when encountering foods, even though they would be fully satiated.

Palatable foods are powerful motivators. Just glancing at a delicious pizza in a restaurant or smelling freshly-baked cookies in the workplace cafeteria may trigger a strong urge for eating. In line with this, viewing appetizing versus bland foods reliably engaged the key components of the brain circuit modulating food anticipation (ventral striatum, thalamus and hypothalamus, and amygdala). Additional activations were found in the posterior cingulate cortex, hippocampus, and ventro-occipital visual regions. These findings are in line with prior work showing that merely seeing cues associated with foods engages the brain's reward circuit[15,17]. Our main new finding was that MOR availability in the thalamus was negatively associated with these visually triggered reward responses (as measured by BOLD-fMRI) in the orbitofrontal and anterior, middle, and posterior cingulate cortices, and ventral striatum. Thus, the more receptors the subjects had, the smaller were their BOLD-fMRI reward responses in these areas.

The association between MOR availability and the BOLD-fMRI reward responses were specific to the reward value of the foods. These associations were seen only with viewing appetizing versus bland foods, and not when viewing the non-food objects (cars). Our results thus reflect reward-specific opioidergic encoding of the seen objects, rather than general opioidergic modulation of food viewing or visual processing in general.

Opioid receptors and peptides are abundantly expressed in the human reward and reinforcement circuit[23] and the endogenous opioid system is involved in both motivating feeding and triggering pleasurable sensations upon food consumption[24]. Our data accord with the work showing that opioid antagonists prevent food-seeking and binge-like eating[7,25]. Furthermore, inverse MOR agonists decrease the hedonia associated with taste and eating in humans[9]. Blockage of the opioid receptors may thus render the opioidergic component of the reward system insensitive to anticipatory food cues, which can consequently inhibit urges to eat when, for example, passing by a cafeteria. In line with this, blocking MOR blunts reward responses to pictures of appetizing foods subjects with binge-eating disorder[25].

Prior fMRI studies have established that viewing food cues such as pictures of appetizing foods engages the striatal reward circuit and amygdala, but also medial and orbitofrontal areas[15,20,26]. We did not observe significant frontocortical responses to the appetizing versus bland foods in the main analyses; however, we found these responses to be associated with MOR availability. The orbitofrontal cortex encodes primary

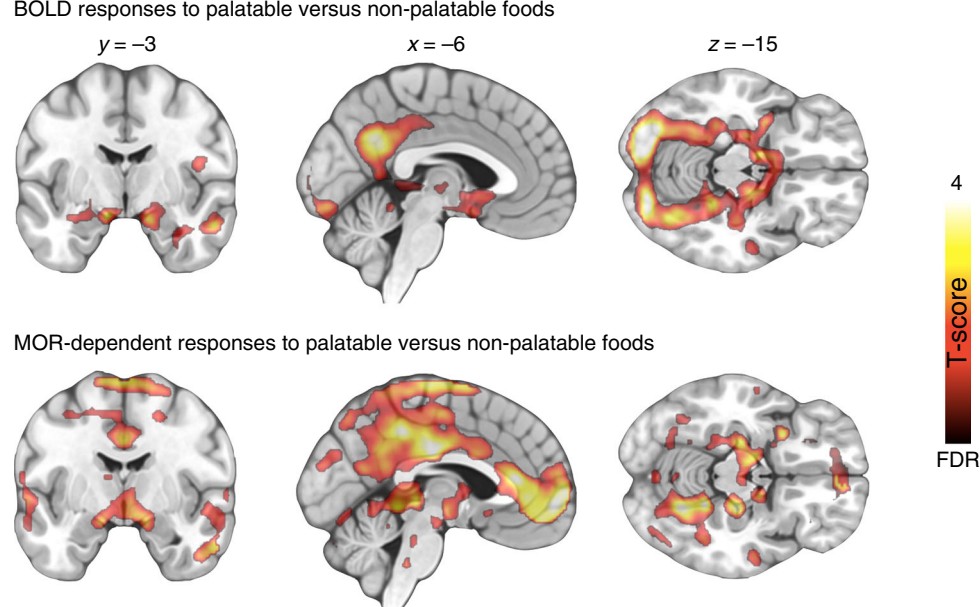

**Fig. 4** BOLD responses to food pictures. Top row: Brain regions where BOLD-fMRI responses were larger when viewing palatable versus non-palatable foods (*n* = 33). Bottom row: brain regions where MOR availability in thalamus were negatively associated with BOLD-fMRI responses when viewing palatable versus non-palatable foods (*p* < 0.05, FDR corrected at cluster level). Colourbar shows the T statistic range

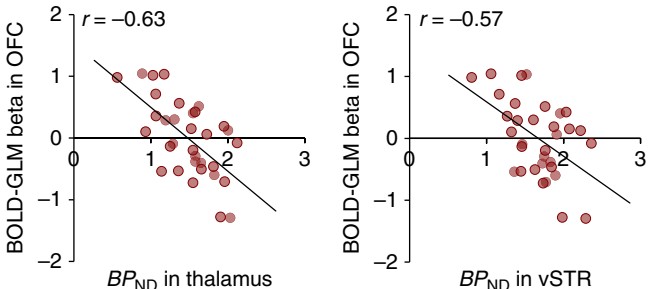

**Fig. 5** Regional association and least-squares regression lines between BOLD-fMRI contrast to palatable minus non-palatable foods in the orbitofrontal cortex (OFC) and [$^{11}$C]carfentanil $BP_{ND}$ in the thalamus and ventral striatum (*n* = 33). The data are plotted for visualization purposes only, and statistical inference is based on the full-volume analysis

reinforces, such as taste, and is involved in controlling reward-related behavior and learning associations between rewards and other sensory signals[27]. Relatedly, it has been proposed that incentive sensitization might be an important risk factor for weight gain and obesity[28]. Accordingly, BOLD responses to appetizing food images in the nucleus accumbens predict the likelihood of future weight gain at least in the short term[29]. Our data suggest that opioidergic pathways may contribute significantly to such anticipatory reward sensitization and consequently development and maintenance of obesity. Consequently, behavioral treatments such as physical exercise could also contribute to reduced feeding and weight loss also due to their capability to engage MORs[30], rather than by merely increasing energy consumption.

Prior PET studies have established alterations in both reward processing and MOR expression in morbid obesity. Overweight subjects have lowered MOR concentration throughout the brain[10,12,13] and elevated anticipatory reward responses while viewing food cues[18–20]. The present study directly links these molecular and functional lines of evidence by showing that healthy subjects who have lowest endogenous MOR levels (and

are in this sense most similar to obese subjects) show the highest anticipatory reward responses, similarly as obese subjects. Consequently, low MOR levels might constitute a risk factor for overeating and weight gain: Low MOR tone could predispose individuals to engaging in food consumption whenever food cues are encountered, and the repeated overstimulation of MOR occurring due to feeding[10] could lead to further downregulation of MOR[12]. This would thus lead to a vicious circle where progressively lowering MOR tone makes the individual increasingly sensitive to food-related cues. Because the theoretical[28] and clinical[10,12,13] importance of the presently observed contribution of MOR to visual reward processes, future studies need to (i) assess whether MOR availability links to anticipatory reward processing in obesity, and (ii) whether alterations in MOR availability and concomitant alterations in reward processing may constitute risk factors for gaining weight.

We studied only young non-obese males, thus our results may not directly generalize to females and other age and weight groups. In addition, our study was strictly related to visual reward processing and no actual rewards were delivered in the study. Whether these results also translate to reward consumption phase remains to be tested. The experiment only involved single reward type (foods), thus we cannot be certain whether the results generalize to other reward types. However, given that multiple reward types activate overlapping limbic areas in both fMRI[31] and PET[2,11,32], it is likely that the opioid system could contribute to reward responses independently of the specific reward category. This, however, needs to be established in future studies. Also, regional MOR availabilities are highly correlated[33], thus it was not possible to disentangle the specific effect of MOR availability in distinct regions on the BOLD responses. However, the association of these global, individual differences in MOR levels yield regionally selective effects when used to predict with the BOLD-fMRI responses, importantly these effects were also functionally selective as they are seen only for the appetizing versus bland contrast.

The PET outcome measure used in the study ($BP_{ND}$) cannot distinguish between receptor density and affinity. Thus, the [$^{11}$C]carfentanil $BP_{ND}$ may reflect either the number of receptor

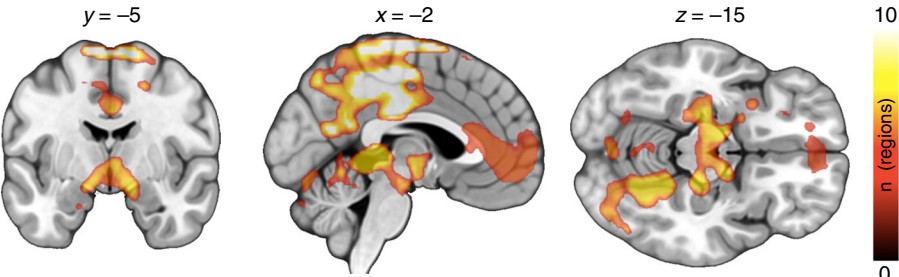

**Fig. 6** Cumulative map showing the number of ROIs (out of 10) whose [$^{11}$C]carfentanil $BP_{ND}$ was correlated ($p < 0.05$, FDR corrected at cluster level) with the voxel-wise BOLD responses to palatable versus non-palatable foods in each brain area ($n = 33$)

proteins or affinity to bind this radioligand. However, short-term reproducibility for this measurement is excellent[34], suggesting that time-dependent fluctuations of baseline endogenous opioid levels do not significantly interfere with $BP_{ND}$ measurements.

We conclude that endogenous MOR availability is associated with anticipatory reward responses when seeing palatable versus non-palatable foods. Given the intimate associations between the MOR system and hedonic and motivational processing of food[24] and the data on MOR downregulation in obesity[10,12,14], these data suggest that low MOR availability may be a risk factor for developing obesity by increasing responsivity and possibly appetitive motivation when encountering food cues.

## Methods

**Subjects**. The ethics committee (EC) of the South-western Finland hospital district approved the study protocol, and the study was conducted in accordance with the Declaration of Helsinki. All subjects signed EC-approved informed consent forms. Thirty-three healthy male adults (age range 19–36, $M_{age} = 25$ years, $SD_{age} = 5$ years) volunteered for the study. We studied only young, non-obese (19.2–26.9 kg/m²; $M_{BMI} = 23$ kg/m², $SD_{BMI} = 2$ kg/m²) males because previous PET studies have established that obesity, age, and sex influence MOR density, as well as the capacity for activating the MOR system[10,12,35–37]. Older subjects tended to have slightly higher BMIs ($r = 0.43$, $p < 0.05$). Exclusion criteria (in addition to standard PET and MRI exclusion criteria) were poor compliance, smoking, excessive alcohol consumption (> 8 U/week), use of illicit drugs, current medication affecting the central nervous system, or a history of current neurological or psychiatric disease confirmed using the structured clinical interview for DSM-IV, medical history, and blood tests. Time was compensated for all subjects. One additional subject's PET data were lost due to technical problems in data acquisition.

**PET data acquisition and analysis**. We measured MOR availability with the agonist radioligand [$^{11}$C]carfentanil that has high affinity for MORs[38]. Radioligand synthesis has been described previously[12,34]. Subjects fasted for at least 2 h before scanning. Data were acquired with the Philips Ingenuity PET-MR scanner and GE Healthcare Discovery 690 PET/CT scanner at Turku PET Centre. After an intra-venous radioligand bolus-injection ($M = 254$ MBq, $SD = 14$ MBq), cerebral radioactivity was measured for 51 min. Self-reported hunger levels were measured on a VAS ranging from 0 to 100 at the beginning of the scans ($M = 33.85$, $SEM = 3.07$). Data were corrected for dead-time, decay, and measured photon attenuation. The dynamic PET scans were reconstructed using the MRP reconstruction method[39].

The data were aligned and co-registered using SPM12 (www.fil.ion.ucl.ac.uk/spm/) and in-house Matlab toolboxes. Head motion was corrected by realigning the dynamic PET images frame-to-frame. Subject-wise T1-weighted MR images were co-registered to the sum image of the realigned PET frames. Reference regions were drawn on T1 images with PMOD 3.3 software (PMOD Technologies Ltd., Zurich, Switzerland). Receptor availability was expressed as $BP_{ND}$—the ratio of specific to non-displaceable radioligand binding. Occipital cortex was used as the reference region, as it is practically devoid of MOR[40]. We next calculated $BP_{ND}$ for each voxel using the simplified reference tissue model (SRTM) using reference tissue time activity curves as input[41]. Prior studies have validated that such outcome measure is not confounded with perfusion or tracer transport[42]. Subject-wise T1 images were first normalized to MNI space (Montreal Neurological Institute (MNI)—International Consortium for Brain mapping), and the resulting warps were subsequently applied to the parametric $BP_{ND}$ images. Finally, the images were smoothed with a 8-mm FWHM Gaussian kernel.

We generated anatomical regions of interest (ROIs) in key components of the reward and emotion circuits (ventral striatum, caudate nucleus, putamen, amygdala, thalamus, insula, orbitofrontal cortex, and anterior, middle, and posterior cingulate cortex) using the AAL[43] and Anatomy[44] toolboxes. Regional

binding in potentials in each ROI were correlated with BMI to test whether MOR availability would be dependent on BMI in the normal weight range similarly as has been previously, and used in the PET-fMRI fusion analysis (see below).

**Experimental design for fMRI**. Experimental design for fMRI is summarized in Fig. 1. The stimulus pictures were color photographs of palatable foods (e.g., cookies, pizza), non-palatable foods (e.g., lentils, bread), and cars, the latter serving as a non-edible, neutral object category. Stimulus categories were matched in terms of low-level visual including mean luminosity, RMS contrast, and global energy. Normative ratings from our prior study[20] show that the palatable foods were considered as more pleasant than the non-palatable foods, $t(31) = 4.67$, $p < 0.001$, or cars, $t(31) = 2.76$, $p = 0.01$, but non-palatable foods and cars did not differ with respect to pleasure ratings, $t(31) = 0.41$.

The stimuli were presented in a conventional box-car design with alternating 15.75 s epochs with six stimuli from one category (palatable foods, non-palatable foods, and cars) shown in each epoch. The stimuli were shown for 1 s each and intermixed with a fixation cross shown for a jittered period (0.75–1.75 s; Mean = 1.25 s). The pictures were shown slightly to the left or to the right of the fixation. Subjects' task was to view the pictures passively, ignore their content, and indicate the picture displacement (left or right) with ipsilateral response button press. This ensured that subjects paid attention to the stimuli while not volitionally judging their reward value. Thus, the results reflect the stimulus-driven, extrinsic responses to the food cues, rather than their intrinsic, volitional evaluation. Because repeated viewing of food pictures could result in carryover effects, each food picture epoch (either palatable or non-palatable foods) was followed by a car picture epoch. Stimulus order was randomized for each epoch. Altogether the subjects saw 72 appetizing food pictures (in 12 epochs), 72 bland food pictures (in 12 epochs), and 144 car pictures (in 24 epochs). The starting epoch of the task was counterbalanced across participants. Altogether the fMRI experiment lasted for 14 min.

**fMRI data acquisition**. Data were acquired with a 3-Tesla Philips Ingenuity PET-MR scanner at Turku PET Centre. Subjects fasted at least 2 h prior to scanning. Functional data were acquired with echo-planar imaging (EPI) sequence, sensitive to the BOLD signal contrast with the following parameters: TR = 2000 ms, TE = 20 ms, 90° flip angle, 240 mm FOV, 80 × 80, 53.4 kHz bandwidth, 3 × 3 × 4 mm voxel size. Each volume consisted of 35 interleaved slices acquired in ascending order without gaps. A total of 430 functional volumes were acquired. Anatomical reference images were acquired using a T1-weighted sequence with 1 mm³ resolution (TR 8.1 ms, TE 3.7 ms, flip angle 7°, scan time 263 s).

Data were preprocessed and analyzed using SPM12 software (www.fil.ion.ucl.ac.uk/spm/). The EPI volumes were first were sinc interpolated in time to correct for slice time differences. Head motion was corrected by realigning the images to the first scan by rigid body transformations. Echo-planar and structural images were co-registered, the structural images was normalized to the MNI space using linear and non-linear transformations, and the resulting warps were subsequently applied to the functional images. Finally, the functional images were smoothed with a 8-mm FWHM Gaussian kernel.

**Analysis of regional effects**. The data were analyzed with a two-stage whole-brain random effects. This assesses effects on the basis of inter-subject variance, and thus allows inferences about the population that the participants were drawn from. First, we used a subject-wise GLM to assess regional effects of the visual stimulation on BOLD responses. The model included all the three experimental conditions (appetizing foods, bland foods, and cars) modeled with box-car functions. Realignment parameters were included as nuisance covariates to account for motion-related variance. High-pass filtering (cut-off 128 s) and AR(1) modeling of temporal autocorrelations was also applied. For each subject, we then generated contrast images for appetizing minus bland foods. These images were then subjected to a second-level analysis for population-level inference. Due to recent concerns regarding the false-positive rates in parametric statistical inference in neuroimaging[45], we also used complementary nonparametric inference with SnPM13 toolbox (http://warwick.ac.uk/snpm; 5000 permutations $p = 0.05$, FWE-corrected at cluster level).

**Fusion analysis of PET and fMRI data**. Mean subject-wise $BP_{ND}$ were extracted for each PET ROI, and used in a full-volume linear regression analysis to predict the voxel-wise contrast estimates (that is, SPM contrast images) for the appetizing versus bland foods contrast. Because of high between-regions co-dependency of [$^{11}$C]carfentanil binding potentials[33,46], each ROI was used as a predictor in separate model. Due to high spatial autocorrelation in the [$^{11}$C]carfentanil $BP_{ND}$[33], we also calculated within-subject mean binding potential across the whole brain to index global MOR availability and used that to predict BOLD responses similarly as described above for the individual ROIs.

**Data availability**. Second-level statistical maps are stored in the NeuroVault database (https://neurovault.org/collections/CQREBTPO/). The PET and MRI brain scans were considered to be sensitive medical information that the ethical board approving the study protocol, and consequently the board did not permit their further distribution. Other datasets and resources are available upon request from the corresponding author.

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

## Acknowledgements

The study was conducted within the Finnish Centre of Excellence in Cardiovascular and Metabolic Diseases and supported by the Academy of Finland (grants #304385, #283320, and #251125 to LN). We thank the staff of Turku PET Centre for their help with data collection.

## Author contributions

L.N. designed the experiments, analyzed the data, and wrote the paper; T.S. acquired and analyzed the data and wrote the paper; L.T., J.H., P.N., and K.K. designed the experiments and wrote the paper; and J.T. acquired and analyzed the data.

## Additional information

**Competing interests:** The authors declare no competing interests.

