## [Peer Review File · Nature Communications]

Reviewers' comments:

Reviewer #1 (Remarks to the Author):

This manuscript examines the relationship between μ -opioid receptor availability and BOLD fMRI responses to appetitive vs. lesser (bland foods) appetitive or non-food stimuli (cars). The authors examined these processes using PET-MRI with what appear to be simultaneous acquisitions. They observed fMRI activation in response to appetitive stimuli, and correlations between both regional and global μ -opioid receptor availability.

The main criticism of this manuscript is that the μ -opioid receptor availability measures as acquired, are a composite of activation of that neurotransmitter system in response to appetitive stimuli, as well as basal levels of receptor availability. Activation of this neurotransmitter system in response to salient or appetitive stimuli would be expected to be positive correlated to the reward value of that stimuli. Here the authors find the exact opposite relationship, consistent with the likelihood that the measures employed are both a result of activation as well as receptor availability (which is decreased with activation). The fact that global measures of receptor binding were also associated with BOLD-fMRI reduce the possibility that the effects observed are regional and specific to food-associated appetitive responses. The lack of relationships with the BOLD-fMRI contrast of appetitive vs, non-food (cars) stimuli seem to point in the same direction.

It would have been helpful if the authors had examined receptor binding in the specific regions activated in the fMRI contrasts, increasing the probability that the effects observed, which nevertheless remain purely correlational, were related to regional μ -opioid receptor effects, even when those represent a composite of receptor availability and endogenous opioid release.

Reviewer #2 (Remarks to the Author):

Here they examine whether MOR availability changes with anticipatory reward responses to visual food cues in non-obese individuals. They found that lower MOR availability in the thalamus is associated with stronger BOLD responses to anticipatory reward in the frontal and cingulate cortices, striatum and amygdala, regardless of BMI. This is an interesting paper and addresses an important question on the role of MOR availability associated with palatable food seeking.

In the introduction, state the hypothesis tested more clearly. "ie We hypothesize that individual differences in MOR occupation mediate anticipatory reward processing and urge to eat. They could remove the "we find that..." paragraph from the introduction as this is repeated in the first paragraph of the results. This will allow more room for description of the results in the results section.

In the methods, they should add some more details on how they tested if their palatable foods were significantly different in desirability to 'bland foods' and cars? Was this assessed prior to fMRI / PET scanning? Was this rating of pleasantness in the same individuals that underwent the fMRI /PET scanning?

They demonstrate a negative association with MOR availability in the amygdala and BMI. Was this the same for ventral striatum? Give some rationale why amygdala was the only region assessed.

They show that MOR availability in the thalamus was negatively associated with BOLD signals for food cues in OFC, cingulate cortex and ventral striatum. However, in their abstract, intro and first paragraph of discussion they suggest that it is MOR availability in the 'reward circuit' that is

negatively correlated with anticipatory feeding. The thalamus is not traditionally considered to be part of the reward circuit per se. Perhaps they could reword this.

In the test for MOR availability in responses to food vs non-food (ie cars) it would be more interesting to do an additional analysis for palatable food vs other non-food rewards (ie money, attractive face or some other predetermined 'favorite thing'). It is the MOR availability in these regions associated with hedonic foods per se or just rewarding or salient things in general.

They should present a supplemental table indicating their raw results of each brain region tested and the correlation coefficient.

Spelling error in Fig 5 legend (purposis = purposes)

Spelling error (cares = cars) – methods – Experimental design for MRI

Reviewer #3 (Remarks to the Author):

Nummenmaa and colleagues carried out complementary studies examining PET imaging of ¹¹C-Carfentanil binding fMRI measurements of brain responses to appetising and bland food images and to non-food images (cars). Studying 33 healthy men with BMIs ranging from lean to overweight, and related mu opioid receptor binding potential to the neural responses relatively specific to appetising food images in key regions. They report that, as well as a negative relationship between MOR BP in amygdala and subjects' BMI there were consistent negative relationships between the magnitude of (appetising vs bland) food picture response and BP across a range of key reward-related regions. They conclude that endogenous MOR availability is associated with anticipatory reward responses and go on to suggest that low MOR availability may be a risk factor for developing obesity by increasing responsivity and possibly appetitive motivation when encountering food cues.

This is an interesting study and the findings are well and clearly presented here. I have a number of concerns about the generalizability of the findings and the interpretability of the findings that make me think that this study does not necessarily support the inferred relationship between opioid receptor function and anticipatory reward responding. My main concern is that there is no real consideration of the mental state of participants viewing the images. The authors refer to this as "anticipatory" responding but this term means something rather specific and implies the imminent receipt of a reward (as, for example, when a stimulus signals that a food reward will be available in, say the next, 2-3 seconds). Simply looking at food images does not necessarily entail any anticipation and, indeed, under the current study context, it seems unlikely that they would be anticipating anything other than the arrival of the subsequent images or blocks of images. The authors also use the term "implicit" with respect to the processing of particular food stimulus attributes, which they argue is the case because the participants were engaged in a task that was unrelated to image content. However, the use of the term "implicit" suggests that image content processing was below the level of awareness which was not necessarily the case and it may well have been that participants (to a greater or lesser degree varying across individuals), processing and attending to the image content as well as carrying out the required task. A more accurate term would be "incidental" rather than implicit perhaps and, in any case, this is another instance in which the lack of a precise understanding of what was going on in the fMRI task puts major constraints on interpretability.

A further concern is that there is no evidence presented about whether people with differing BMI/age etc found the appetising and bland images rewarding to differing degrees. Again, absence of this understanding means that one cannot rule out the possibility that BP correlated not with anticipatory responding but with, say, liking or valuation (an effect that could itself be moderated or mediated by BMI)

Further questions:

Did age correlate with BMI? Were both entered as regressors in the correlation models?

Was there a correction for all of the regional correlations measured

Were participants fasted? Did they differ on hunger and satiety ratings pre-scanning?

Response to Reviewers' comments:

Reply to Reviewer #1

The main criticism of this manuscript is that the μ -opioid receptor availability measures as acquired, are a composite of activation of that neurotransmitter system in response to appetitive stimuli, as well as basal levels of receptor availability. Activation of this neurotransmitter system in response to salient or appetitive stimuli would be expected to be positive correlated to the reward value of that stimuli. Here the authors find the exact opposite relationship, consistent with the likelihood that the measures employed are both a result of activation as well as receptor availability (which is decreased with activation).

Authors' response

Baseline [11C]-carfentanil BP_{ND} was measured separately from the task fMRI and therefore it is unlikely that the response to the stimuli would directly affect the BP_{ND} – no MOR activation measures were thus obtained. However, we acknowledge that baseline [11C]-carfentanil BP_{ND} represents a composite measure of receptor density and affinity. Short term reproducibility for this measurement is excellent (Hirvonen et al., 2009), suggesting that time-dependent fluctuations of baseline endogenous opioid levels do not significantly interfere with BP_{ND} measurements. Rather, baseline mostly represent receptor density (Hietala, Nagren, Lehtikoinen, Ruotsalainen, & Syvalahti, 1999) Unlike receptor affinity states or occupancy by endogenous opioids, receptor density is both genetically determined and subject to long term regulation via altered protein synthesis, *i.e.*, in the time frame from days to weeks. Thus, baseline BP_{ND} in part represents the long term regulatory state of endogenous opioid tone. Downregulation, or reduction of receptor density via lower protein synthesis, is a common desensitization mechanism for G-protein coupled receptors, including the mu opioid receptor. We have previously argued that, for example, downregulation of mu opioid receptors in morbidly obese subjects, represented by lower [11C]-carfentanil BP_{ND} , is due to previous overstimulation by endogenous opioids, potentially elicited by overeating (Karlsson et al., 2016; Karlsson et al., 2015). This phenomenon might explain the current finding of a negative correlation between anticipatory food reward and [11C]-carfentanil BP_{ND} . That is, subjects who show higher visual reward may have lower receptor density as a compensatory desensitization phenomenon due to overstimulation of MOR by endogenous opioids.

The fact that global measures of receptor binding were also associated with BOLD-fMRI reduce the possibility that the effects observed are regional and specific to food-associated appetitive responses. The lack of relationships with the BOLD-fMRI contrast of appetitive vs, non-food (cars) stimuli seem to point in the same direction.

Authors' response

The reviewer is correct in that the PET measures of mu-opioid levels lack regional selectivity, but this is simply due to i) mu receptors being expressed abundantly in the human brain and ii) the tight interregional coupling of mu receptor levels within individual brains (Karjalainen et al., 2017; Tuominen, Nummenmaa, Keltikangas-Jarvinen, Raitakari, & Hietala, 2014). That is, an individual having above-average number of mu receptors in amygdala has also above-average number of mu receptors in thalamus

and elsewhere in the brain. However, the association of the individual differences in μ levels yield *regionally selective* effects when used to predict with the BOLD-fMRI responses. Importantly, these effects are also *functionally selective* as they are seen only for the appetizing versus bland foods contrast. We have now added a clarification regarding the global μ effects on page 9:

“Also, regional MOR availabilities are highly correlated (Tuominen, et al., 2014), thus it was not possible to disentangle the specific effect of MOR availability in distinct regions on the BOLD responses. However, the association of these global, individual differences in μ levels yield regionally selective effects when used to predict with the BOLD-fMRI responses, importantly these effects were also functionally selective as they are seen only for the appetizing versus bland contrast.”

It would have been helpful if the authors had examined receptor binding in the specific regions activated in the fMRI contrasts, increasing the probability that the effects observed, which nevertheless remain purely correlational, were related to regional μ -opioid receptor effects, even when those represent a composite of receptor availability and endogenous opioid release.

Authors' response

We would like to clarify that our approach was based on a priori definition of the regions of interest in the reward and emotion circuits for the PET data. Rather than looking for co-localization of receptor density and fMRI responses, we were interested in the modulatory role of the opioid circuit in the reward responses, possibly in the reward circuit but also elsewhere in the brain. The results of these analyses are shown in **Figure 4**; for similar approach see e.g. (Karjalainen, et al., 2017). Regarding endogenous opioid release, please see our first reply to Reviewer #1 – our measurements pertain to receptor density rather than endogenous neurotransmitter release.

Reply to Reviewer #2

In the introduction, state the hypothesis tested more clearly. “ie We hypothesize that individual differences in MOR occupation mediate anticipatory reward processing and urge to eat. They could remove the “we find that...” paragraph from the introduction as this is repeated in the first paragraph of the results. This will allow more room for description of the results in the results section.

Authors' response

Noted and changed as suggested on **page 3:**

“Based on prior observations on hypoactive reward responses to food cues in the obese (Nummenmaa et al., 2012; Rothmund et al., 2007; Stoeckel et al., 2008), endogenous opioid release following feeding (Tuulari et al., 2017), and lower MOR density as a compensatory desensitization phenomenon due to overstimulation of MOR by endogenous opioids (Burghardt, Rothberg, Dykhuys, Burant, & Zubieta, 2015; Karlsson, et al., 2016; Karlsson, et al., 2015) we expected to see largest BOLD-fMRI reward responses in subjects with lowest MOR availability.”

In the methods, they should add some more details on how they tested if their palatable foods were significantly different in desirability to ‘bland foods’ and cars? Was this assessed prior to fMRI / PET scanning? Was this rating of pleasantness in the same individuals that underwent the fMRI /PET scanning?

Authors' response

This stimulus set has been used in numerous previous behavioural and imaging studies (e.g. Nummenmaa, Hietanen, Calvo, & Hyona, 2011; Nummenmaa, et al., 2012; Passamonti et al., 2009). Because the subjective ratings between the appetizing and bland foods are very consistent across subjects, we did not obtain individual ratings in this study as the protocol was already very demanding with both PET and MRI scans. The ratings shown are from our earlier study (Nummenmaa, et al., 2012), this is now stated in the manuscript **page 10.**

They demonstrate a negative association with MOR availability in the amygdala and BMI. Was this the same for ventral striatum? Give some rationale why amygdala was the only region assessed.

Authors' response

ROI-level analyses were conducted for all the a priori PET ROIs. The association was significant (in two-way testing) in amygdala, $r = -0.39$, $p < 0.05$. In one-way testing, significant association was also observed in thalamus, putamen, and ventral striatum ($r_s < .029$, $p_s < 0.05$). This has been amended on **pages 4 and 10.** We acknowledge that the corresponding p-values were not subjected to multiple comparison correction, but think that the associations should nevertheless be reported because i) correlation coefficients can essentially be interpreted as effect sizes independently of significance testing and ii)

because these observations were in line with our predictions and prior data on obesity and MOR availability (Burghardt, et al., 2015; Karlsson, et al., 2015).

They show that MOR availability in the thalamus was negatively associated with BOLD signals for food cues in OFC, cingulate cortex and ventral striatum. However, in their abstract, intro and first paragraph of discussion they suggest that it is MOR availability in the ‘reward circuit’ that is negatively correlated with anticipatory feeding. The thalamus is not traditionally considered to be part of the reward circuit per se. Perhaps they could reword this.

Authors’ response

The reviewer is correct – we have now reworded this as “reward and emotion circuit”

In the test for MOR availability in responses to food vs non-food (ie cars) it would be more interesting to do an additional analysis for palatable food vs other non-food rewards (ie money, attractive face or some other predetermined ‘favorite thing’). It is the MOR availability in these regions associated with hedonic foods per se or just rewarding or salient things in general.

Authors’ response

We agree that it would have been interesting to parse the effects of mu receptors on different types of reward signals. Unfortunately, we did not consider that when initiating the study; as this was the first-ever PET-MRI study comparing MOR availability with fMRI reward responses, we decided to maximize the statistical power and focus on single, powerful reward signal (food pictures). We have now described this on **pages 8-9**

“The experiment only involved single reward type (foods) thus we cannot be certain whether the results generalize to other reward types. However, given that multiple reward types activate overlapping limbic areas in both fMRI (Noori, Cosa Linan, & Spanagel, 2016) and PET (Manninen et al., 2017; Nummenmaa & Tuominen, 2017; Tuulari, et al., 2017), it is likely that the opioid system could contribute to reward responses independently of the specific reward category, yet this needs to be established in future studies.

They should present a supplemental table indicating their raw results of each brain region tested and the correlation coefficient.

Authors’ response

In our view activation tables are not an optimal way for presenting the neuroimaging data, as the cluster peak coordinates may be misleading with respect to the actual area covered by the cluster, particularly if the cluster is large. We have thus uploaded the results files to NeuroVault (<https://neurovault.org/collections/CQREBTPO/>) where their inspection is convenient. If the Editor and Reviewer consider the Tables essential, we can provide them too.

Spelling error in Fig 5 legend (purposis = purposes)

Spelling error (cares = cars) – methods – Experimental design for MRI

Authors' response

Noted and changed

Reply to Reviewer #3

This is an interesting study and the findings are well and clearly presented here. I have a number of concerns about the generalizability of the findings and the interpretability of the findings that make me think that this study does not necessarily support the inferred relationship between opioid receptor function and anticipatory reward responding. My main concern is that there is no real consideration of the mental state of participants viewing the images.

Authors' response

The reviewer is correct in the sense that we did not control for subjects' mental state while viewing the images. However, we must note that this type of passive visual reward cue task has been widely used in prior studies (e.g. Goldstone et al., 2009; Nummenmaa, et al., 2012, and many others; Passamonti, et al., 2009). Our goal was to study stimulus-driven (extrinsic) responses, rather than subjective (intrinsic) evaluation of the stimuli, and we wanted our paradigm to be comparable with the substantial prior literature on visual reward processing. This has been clarified on **page 10**:

“The pictures were displaced slightly to the left or to the right of the screen, and the participants were instructed to view the pictures passively, ignore their content and press the left or right response button according to which side the stimulus was presented. This ensured that subjects paid attention to the stimuli while not volitionally judging their reward value, thus we could focus on the stimulus-driven, extrinsic responses to the food cues.”

The authors refer to this as “anticipatory” responding but this term means something rather specific and implies the imminent receipt of a reward (as, for example, when a stimulus signals that a food reward will be available in, say the next, 2-3 seconds). Simply looking at food images does not necessarily entail any anticipation and, indeed, under the current study context, it seems unlikely that they would be anticipating anything other than the arrival of the subsequent images or blocks of images.

Authors' response

The reviewer is correct that this terminology, albeit used in this type of studies, is not strictly speaking correct even though in natural feeding situations, seeing a food predicts that feeding-induced reward is likely (if the individual decides to eat) and that viewing pictures of foods is indeed associated with pleasure. Consequently, we have revised the terminology so that “anticipatory reward processing” has been replaced with “visual reward” to better reflect the fact that i) the exact reward phase cannot be determined and ii) no food-related reward was delivered in the experiment.

The authors also use the term “implicit” with respect to the processing of particular food stimulus attributes, which they argue is the case because the participants were engaged in a task that was unrelated to image content. However, the use of the term “implicit” suggests that image content processing was below the level of awareness which was not necessarily the case and it may well have been that participants (to a greater or lesser degree varying across individuals), processing and attending to the image content as well as carrying out the required task. A more accurate term would be “incidental”

rather than implicit perhaps and, in any case, this is another instance in which the lack of a precise understanding of what was going on in the fMRI task puts major constraints on interpretability.

Authors' response

We have now removed the term “implicit” from the description of the fMRI method. We would also like to clarify that in our task subjects were not engaged in volitional judgment of the pictures' reward value. This does not mean that the pictures would have failed to result in conscious percept, and we do not equate implicit with lack of awareness. Instead of volitionally judging whether the pictures show appetizing or bland foods, the subjects were asked to ignore the picture content and simply judge whether the pictures were shown on left or right of the fixation. This of course does not mean that the participants would not have been able to evaluate the stimuli volitionally if they wanted to do so (Lahteenmaki, Hyona, Koivisto, & Nummenmaa, 2015). Our main argument however is not that our results reflect processing of unattended or subliminal food cues. Instead, our task was designed to mimic the situation where an individual encounters the omnipresent food advertisements, which most of the time go unnoticed as they are not directly relevant to the current task at hand. The method section has been updated on **page 10**:

“The pictures were displaced slightly to the left or to the right of the screen, and the participants were instructed to view the pictures passively, ignore their content and press the left or right response button according to which side the stimulus was presented. This ensured that subjects paid attention to the stimuli while not volitionally judging their reward value. Thus, the results reflect the stimulus-driven, extrinsic responses to the food cues, rather than their intrinsic, volitional evaluation.”

A further concern is that there is no evidence presented about whether people with differing BMI/age etc found the appetising and bland images rewarding to differing degrees. Again, absence of this understanding means that one cannot rule out the possibility that BP correlated not with anticipatory responding but with, say, liking or valuation (an effect that could itself be moderated or mediated by BMI)

Authors' response

Our main focus was on establishing the link between MOR and visual reward processing, rather than examining the endophenotypes causing alterations in this coupling. Consequently, we scanned young males ($M = 25.5$, $SD = 4.9$) with a narrow non-obese BMI range ($M = 23.4$, $SD = 1.7$). Unfortunately, we do not have subjectwise ratings of the food images, but age and hunger levels were not associated MOR availabilities, additionally including age, hunger levels, and weight as nuisance covariates did not alter the BOLD-fMRI responses.

Did age correlate with BMI?

Authors' response

Even though the ranges of age and BMI were narrow, they were correlated ($r = 0.43, p < 0.05$). This is now mentioned on page 9.

Were both entered as regressors in the correlation models?

Authors' response

As the subjects were in a narrow non-obese BMI range ($M = 23.4, SD = 1.7$) and young adults ($M = 25.5, SD = 4.9$) we did not initially include them as covariates as we did not expect to find significant effects in this homogenous sample. We have however replicated the fMRI analyses using age and BMI as covariates of no interest, the results remain essentially unchanged. This is now reported on page 4

Was there a correction for all of the regional correlations measured

Authors' response

As stated on page 11 the analyses were corrected using cluster-level FDR, and also validated using permutation-based tests.

Were participants fasted? Did they differ on hunger and satiety ratings pre-scanning?

Authors' response

The subjects fasted for at least two hours before the scan. This is explained on pages 9 and 10. Hunger levels were measured using a VAS scale ranging from 0 to 100. There are individual differences in the self-reported hunger levels, however these were in general relatively small ($SEM = 3.07$). The hunger levels are now reported in page 9.

References

- Burghardt, P. R., Rothberg, A. E., Dykhuis, K. E., Burant, C. F., & Zubieta, J. K. (2015). Endogenous opioid mechanisms are implicated in obesity and weight loss in humans. *Journal of Clinical Endocrinology & Metabolism*, *100*, 3193-3201.
- Goldstone, A. P., de Hernandez, C. G. P., Beaver, J. D., Muhammed, K., Croese, C., Bell, G., . . . Bell, J. D. (2009). Fasting biases brain reward systems towards high-calorie foods. *European Journal of Neuroscience*, *30*, 1625-1635.
- Hietala, J., Nagren, K., Lehtikoinen, P., Ruotsalainen, U., & Syvalahti, E. (1999). Measurement of striatal d-2 dopamine receptor density and affinity with c-11 -raclopride in vivo: A test-retest analysis. *Journal of Cerebral Blood Flow and Metabolism*, *19*, 210-217.

- Hirvonen, J., Aalto, S., Hagelberg, N., Maksimow, A., Ingman, K., Oikonen, V., . . . Scheinin, H. (2009). Measurement of central mu-opioid receptor binding in vivo with pet and [11c]carfentanil: A test-retest study in healthy subjects. *Eur J Nucl Med Mol Imaging*, *36*, 275-286.
- Karjalainen, T., Karlsson, H. K., Lahnakoski, J. M., Glerean, E., Nuutila, P., Jaaskelainen, I. P., . . . Nummenmaa, L. (2017). Dissociable roles of cerebral mu-opioid and type 2 dopamine receptors in vicarious pain: A combined pet-fmri study. *Cereb Cortex*, 1-10.
- Karlsson, H. K., Tuominen, L., Tuulari, J. J., Hirvonen, J., Honka, H., Parkkola, R., . . . Nummenmaa, L. (2016). Weight loss after bariatric surgery normalizes brain opioid receptors in morbid obesity. *Molecular Psychiatry*, *21*, 1057-1062.
- Karlsson, H. K., Tuominen, L., Tuulari, J. J., Hirvonen, J., Parkkola, R., Helin, S., . . . Nummenmaa, L. (2015). Obesity is associated with decreased mu-opioid but unaltered dopamine d-2 receptor availability in the brain. *Journal of Neuroscience*, *35*, 3959-3965.
- Lahteenmaki, M., Hyona, J., Koivisto, M., & Nummenmaa, L. (2015). Affective processing requires awareness. *Journal of Experimental Psychology-General*, *144*, 339-365.
- Manninen, S., Tuominen, L., Dunbar, R. I. M., Karjalainen, T., Hirvonen, J., Arponen, E., . . . Nummenmaa, L. (2017). Social laughter triggers endogenous opioid release in humans. *The Journal of Neuroscience*, *37*, 6125-6131.
- Noori, H. R., Cosa Linan, A., & Spanagel, R. (2016). Largely overlapping neuronal substrates of reactivity to drug, gambling, food and sexual cues: A comprehensive meta-analysis. *European Neuropsychopharmacology*, *26*, 1419-1430.
- Nummenmaa, L., Hietanen, J. K., Calvo, M. G., & Hyona, J. (2011). Food catches the eye but not for everyone: A bmi-contingent attentional bias in rapid detection of nutriments. *Plos One*, *6*, 7.
- Nummenmaa, L., Hirvonen, J., Hannukainen, J. C., Immonen, H., Lindroos, M. M., Salminen, P., & Nuutila, P. (2012). Dorsal striatum and its limbic connectivity mediate abnormal anticipatory reward processing in obesity. *Plos One*, *7*, 10.
- Nummenmaa, L., & Tuominen, L. J. (2017). Opioid system and human emotions. *British Journal of Pharmacology*.
- Passamonti, L., Rowe, J. B., Schwarzbauer, C., Ewbank, M. P., von dem Hagen, E., & Calder, A. J. (2009). Personality predicts the brain's response to viewing appetizing foods: The neural basis of a risk factor for overeating. *Journal of Neuroscience*, *29*, 43-51.
- Rothmund, Y., Preuschhof, C., Bohnert, G., Bauknecht, H.-C., Klingebiel, R., Flor, H., & Klapp, B. F. (2007). Differential activation of the dorsal striatum by high-calorie visual food stimuli in obese individuals. *NeuroImage*, *37*, 410-421.
- Stoeckel, L. E., Weller, R. E., Cook Iii, E. W., Twieg, D. B., Knowlton, R. C., & Cox, J. E. (2008). Widespread reward-system activation in obese women in response to pictures of high-calorie foods. *NeuroImage*, *41*, 636-647.
- Tuominen, L., Nummenmaa, L., Keltikangas-Jarvinen, L., Raitakari, O., & Hietala, J. (2014). Mapping neurotransmitter networks with pet: An example on serotonin and opioid systems. *Human Brain Mapping*, *35*, 1875-1884.
- Tuulari, J. J., Tuominen, L., de Boer, F. E., Hirvonen, J., Helin, S., Nuutila, P., & Nummenmaa, L. (2017). Feeding releases endogenous opioids in humans. *Journal of Neuroscience*, *37*, 8284-8291.

REVIEWERS' COMMENTS:

Reviewer #1 (Remarks to the Author):

The authors have constructively responded to the reviewers comments and modified the manuscript accordingly.

Reviewer #2 (Remarks to the Author):

The authors have addressed all of my previous concerns.

Reviewer #3 (Remarks to the Author):

As I said in my previous review, the absence of any sort of behavioral or cognitive measures associated with the task used during fMRI, in my view, severely limits the conclusions can be drawn about the relationship between variability in MOR and neurocognitive function. In toning down the specificity of their language to talk about visual reward processing, the authors acknowledge this and they have been very responsive to the reviewers' comments overall.

In the end, I feel that the lack of specificity of the fMRI task severely limits any conclusions and makes this an interesting but preliminary study